# Insight Miner: A Time Series Analysis Dataset for Cross-Domain Alignment with Natural Language

**Yunkai Zhang**\* †
IEOR Department
UC Berkeley
yunkai_zhang@berkeley.edu

**Yawen Zhang**\*
Mineral
yawenz@mineral.ai

**Ming Zheng**
Mineral
zhengming@mineral.ai

**Kezhen Chen**
Mineral
kezhenchen@mineral.ai

**Chongyang Gao**†
Computer Science Department
Northwestern University
Chongyanggao2026@u.northwestern.edu

**Ruian Ge**
IEOR Department
UC Berkeley
ruian_ge@berkeley.edu

**Siyuan Teng**
IEOR Department
UC Berkeley
siyuan_teng@berkeley.edu

**Amine Jelloul**
IEOR Department
UC Berkeley
amine_jelloul@berkeley.edu

**Jinmeng Rao**
Mineral
jinmengrao@mineral.ai

**Xiaoyuan Guo**
Mineral
xiaoyuanguo@mineral.ai

**Chiang-Wei Fang**
IEOR Department
UC Berkeley
chiangwei_fang@berkeley.edu

**Zeyu Zheng**
IEOR Department
UC Berkeley
zyzheng@berkeley.edu

**Jie Yang**
Mineral
yangjie@mineral.ai

## Abstract

Time-series data is essential in various science and industry domains, like environmental analysis, agriculture, transportation, and finance. Researchers need to use their domain knowledge to conduct insight mining from time-series data to study scientific topics. However, this process is time-consuming and highly depends on expert knowledge. This paper proposes a large-scale multimodal model (LMM), Insight Miner, to generate decent and comprehensive time-series descriptions with domain-specific knowledge. To introduce rich time-series insights to Insight Miner, we propose a time-series analysis dataset, TS-Insights, composed of time series and textual insight pairs. In the TS-Insights dataset,

---

\*Equal contribution.
†Work done during internship at Google X - Mineral.ai.

NeurIPS 2023 AI for Science Workshop.

we include 100k time series windows sampled from 20 forecasting datasets spanning a wide variety of domains and granularities. Through a meticulous combination of heuristics and statistical tools, we preprocess each raw time series window and use GPT-4 to generate a coherent trend description based on the extracted features. After training with the TS-Insights dataset via instruct tuning, the Insight Miner model performs better in generating time series descriptions and insights compared with state-of-the-art multimodality models, such as LLaVA [1] and GPT-4. Our findings suggest a promising direction of leveraging LMMs for time series analysis and potentially offering avenues for efficient insight mining in scientific domains. The TS-Insights dataset is available here: https://drive.google.com/drive/folders/1qGXigxE5GvmF1oLuGXaqLMkRgwoQfZ7V?usp=sharing.

# 1 Introduction

Time series data has been widely studied in a wide range of domains. Traditionally, researchers have relied on the statistical tools to analyze time series data. Methods such as autoregressive integrated moving average (ARIMA) [2], seasonal decomposition of time series (STL) [3], and the state space models [4] have long been employed for forecasting, detecting seasonality, and understanding the underlying trends in time series datasets. The use of these techniques have been particularly prevalent in fields like economics, meteorology, and transportation to provide effective interpretation of time series data.

Recently, many studies have explored the usage of LLMs for time-series tasks. For example, there are studies leveraging the pretrained LM (GPT2 model) for various time-series tasks (forecasting, classification, anomaly detection, etc.) [5, 6] and achieved the state-of-the-art performance, which demonstrates the universality of pretrained LMs. Another study designed structured prompts to enable zero-shot or few-shot inferences by LLMs [7, 8]. However, the above works mainly focus on tasks where the output is time series or scalars. Directly training LLMs to perform traditional time series tasks such as forecasting or classification does not enable LLMs to handle tasks where the output involves natural language.

On the other hand, the emergence of multimodal LLMs like LLaVA [9] has inspired researchers to investigate approaches to better align domain-specific time-series data with LLM. One such example is the FinVis-GPT [10], which was built on top of the LLaVA model and generated a financial task oriented dataset for alignment and instruction tuning. The proposed FinVis-GPT demonstrates the feasibility of utilizing multimodal LLMs in analyzing financial charts. Our work is also motivated by the success of multimodal LLMs but not limited to a certain domain. We focus on constructing a time series analysis dataset for LMMs. To the best of our knowledge, there is no such dataset for the purpose of aligning time-series data with comprehensive textual descriptions.

In summary, the main contributions of our work are two-fold: 1) we present a time series analysis dataset that enables LLMs to generate faithful time series descriptions, and 2) the proposed dataset is the first large-scale repository that allows time-series data to be aligned into the language embedding space, paving the way for future studies on using large multimodal models to analyze time-series data and provide language insights.

# 2 TS-Insights Dataset

To our knowledge, there are no existing large-scale datasets of time series and language description pairs, let alone for time series analysis. To bridge this gap, we design and generate the first dataset, TS-Insights Dataset, with time series and language pairs for general time series analysis.

Formally, given $N$ time series datasets $\{\mathcal{D}_i\}_{i=1}^N$, where each dataset $\mathcal{D}_i$ has $T_i$ total time steps and $M_i$ features, i.e., $D_i = \{X_j\}_{j=1}^{T_i}$ and $X_j \in \mathbb{R}^{M_i}$, the goal is to generate a question-answer pair for each time series window $W_k \in \mathbb{R}^{m_k \times \tau_k}$ randomly sampled from the $N$ datasets, where $\tau_k$ represents the number of time steps and $m_k$ represents the number of features, which are both randomly subsampled from the chosen dataset.[3] Each training sample consists of a time series window $W_k$, a question $L_k^Q$,

---

[3]To generate the current dataset, $\tau_k$ is randomly sampled from 30 to 500.

and an answer $L_k^A$. Using $(W_k, L_k^Q, L_k^A)$, we create a single-round instruction-following example [1]:

$$\textbf{Human: } W_k \backslash \text{n } L_k^Q < \text{STOP} > \backslash \text{n } \textbf{Assistant: } L_k^A < \text{STOP} > \backslash \text{n}. \qquad (1)$$

To generate such datasets for modalities such as images [1] or biomedical images [11], the common practice is to prompt language-only GPT-4. For example, LLaVA [1] asks GPT-4 to generate multi-turn conversations given the image caption and the bounding boxes of the objects in the image. However, the time series modality presents unique challenges since 1) there are no original captions available for a time series window, 2) existing tools cannot readily convert a time series segment into an input format that is suitable for language-only GPTs, and 3) the semantic meanings of time series windows are more difficult to be described in natural languages.

To address the third challenge, we focus on time series windows that contain a single feature, i.e., $W_k \in \mathbb{R}^{1 \times \tau_k}$, and following traditional time series analysis [12], we generate descriptions based on the trend, the seasonality, and the residuals that are present in the window. A naive solution is to feed in the raw time series as a vector when prompting GPT-4, e.g., "Given the time series [0.52, 0.98, 0.95, 0.91, 1.24, ..., 1.32], generate a description about its trend, seasonality, and volatility." However, we found that GPT-4 fails to accurately extract each component from the raw vector.[4] Instead, we leverage a statistical Seasonal-Trend Decomposition (STL) model to decompose the original time series into a trend component, a seasonality component, and a residual component, and generate a description only based on one component at a time. As a proof of concept, we focus on the trend description in the current version of this paper.

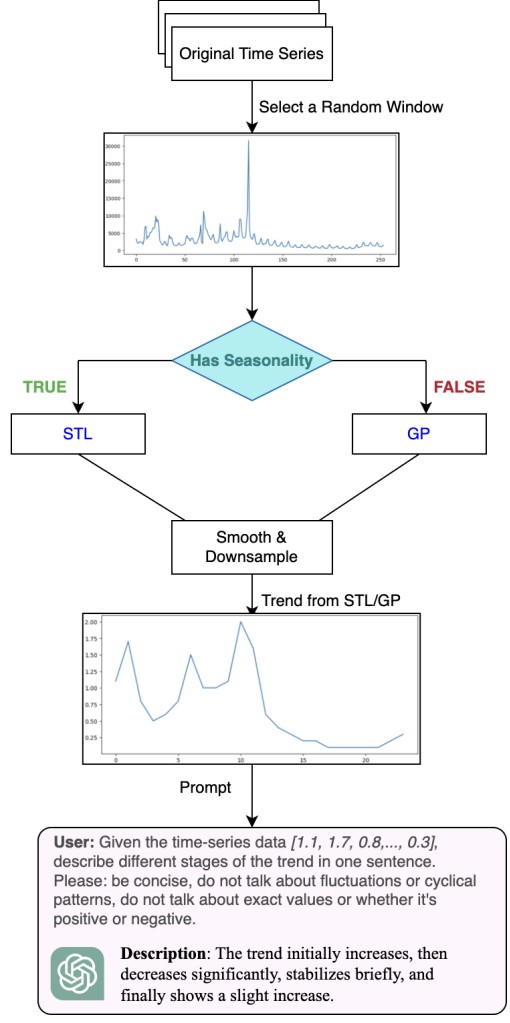

Figure 1: Trend dataset generation workflow.

### 2.1 Trend Generation Workflow

To generate the trend description for a given time series window $W_k \in \mathbb{R}^{1 \times \tau_k}$, we first apply an STL decomposition to extract the trend

$$W_k = \mathcal{T}_k + \mathcal{S}_k + \mathcal{R}_k, \qquad (2)$$

where $\mathcal{T}_k$, $\mathcal{S}_k$, and $\mathcal{R}_k$ denote the extracted trend, seasonality, and residual components, respectively. Denote the value at each individual time step of the extracted trend as $\mathcal{T}_k = (\hat{y}_1, \hat{y}_2, \cdots, \hat{y}_{\tau_k})$.

In some cases, $W_k$ might not have any seasonalities. In such cases, we fit a Gaussian Process (GP) to the $\tau_k$ time steps in the window. Let $W_k = (y_1, y_2, \cdots, y_{\tau_k})$, where $y_i$ is the value at each time step. $W_k$ is modeled by a standard zero-mean GP, whose covariance structure is defined by a kernel $K(.,.)$. Here, the kernel used is a combination of a RBF kernel to model the dependency among different time steps and a white-noise kernel to model the observational noise. That is, $W_k \sim GP(\mu(x), K(x, x'))$, where $\mu(x) = 0$, $K(x, x') = RBF(x, x') + \sigma_e^2 \delta_{x,x'}$, $RBF(x, x') = \sigma_r^2 exp(-\frac{(x-x')^2}{2\gamma})$ and $\delta_{x,x'}$ is the Kronecker delta. The parameters $\sigma_r^2$, $l$ and $\sigma_e^2$ are estimated from the data by maximizing the likelihood. We

---

[4]Examples are shown in Appendix B.

then compute the fitted mean of the Gaussian Process regression at the respective time steps to get $\mathcal{T}_k = (\hat{y}_1, \hat{y}_2, \cdots, \hat{y}_{\tau_k})$ as the extracted trend.

We then apply a Gaussian kernel $\mathcal{F}_k = [\mathcal{F}_1, \mathcal{F}_2, \cdots, \mathcal{F}_{w_k}]$, where $w_k$ is a hyperparameter for the kernel size, to further smooth out the trend, and followed by downsampling with stride size $s_k$[5]:

$$\tilde{y}_i = \sum_{j=-w_k//2}^{w_k//2} \hat{y}_{s_k \cdot i - j} \cdot \mathcal{F}_{w_k//2+j} \tag{3}$$

for $i = 1, 2, \cdots, \tau_k//s_k$.

Finally, we round each entry of $(\tilde{y}_1, \tilde{y}_2, \cdots, \tilde{y}_{\tau_k//s_k})$ to one decimal place and feed it to GPT-4. As such, one data sample pair consists of the original time series window $W_k$ and the trend description generated by GPT-4. An overview of the workflow and the exact prompt we use is shown in Figure 1.

## 2.2 Trend Description Dataset

Using the approach above, we generate 10k samples based on twenty-nine datasets from Monash Time Series Forecasting Archive [13], and leave the other eleven datasets as holdout sets, which are only used for evaluation but not for training. The twenty-nine datasets span a wide range of domains, such as energy [14], weather, traffic [15], and healthcare [16]. Notably, we only sample windows from the train split of each dataset, defined to be the first $70\%$ of the time steps in temporal order. Some datasets contain multiple levels of seasonalities, e.g., daily and weekly. Under the original granularity, each window might not contain enough time steps to discern the higher level of seasonalities, since at least two full cycles are required to conclude there to be a seasonality. As trends should be described after seasonalities are removed, for each dataset, we also aggregate multiple time steps into one time step in order to introduce samples with more diversified patterns.

To further increase the number of training samples in a cost-efficient manner, for each GPT-4 labeled sample pair, we additionally apply nine different random augmentations to the original time series window $W_k$ such that the trend description is still applicable to the augmented samples. We then rephrase the original description generated by GPT-4 using GPT-3.5-turbo in order to increase the language diversity. Therefore, for each original sample, we now have nine augmented samples, resulting in 100k total training samples. A detailed list of test and holdout datasets, the number of samples we generate for each aggregated granularity level, as well as a list of augmentation methods can be found in Appendix A.

## 3 Insight Miner

We use the checkpoint from LLaVA [1], a general-domain vision-language multimodal conversation model as a starting point, and continue finetuning the LLaVA weights to the time series domain. We use the same neural network architecture as LLaVA: we first convert the time series window into an image using lineplot, feed the image into the vision encoder, and then use a linear projection layer to map the vision output into the language embedding space, finally, the language model takes in the projected image embeddings concatenated with the language instructions as the input and returns the language response.

To align the time-series images with the LLM, we only finetune the linear projection layer, while keeping both the vision encoder and the language model frozen. For each training sample, we show the original time series to the model in the form of a line plot and the language instruction is to ask it to describe the trend, and the goal is to predict the description generated by the GPTs. The final model is named Insight Miner.

Note that the training cost of Insight Miner is relatively affordable as it was trained using $8 \times$ A100 40GiB GPUs. Each epoch takes around an hour to train. Once the model finishes training, it can be easily deployed at a low inference cost.

---

[5]We choose stride size $s_k$ so that $\tau_k//s_k = 25$.

## 4  Experiments

We conduct experiments to evaluate how well the trend dataset can enable large multimodal models to generate trend descriptions that are faithful to the original time series. More specifically, we sample 119 total windows for evaluation. Among these, 69 examples are from the test split (last 30%) of the same datasets we used for training, and the other 50 examples are from the holdout datasets which are not used for training entirely. The models we include for comparison are:

- LLaVA [1]: using the checkpoint publicly available on HuggingFace.
- Vision (3 epochs): finetuned from the above LLaVA checkpoint for three epochs using the generated trend dataset. It takes in the original time series window plotted using the lineplot function in the Seaborn package.
- Vision (1 epoch): finetuned from the above LLaVA checkpoint for one epochs using the generated trend dataset.
- Engineering GPT: GPT-4 that takes in the extracted features as described in Section 2.1.

Here, Vision (3 epochs) and Vision (1 epoch) are two versions of our Insight Miner trained using a different number of epochs. As we observed feeding the raw time series vector into GPT-4 leads to inferior descriptions compared to Engineering GPT, we do not include it for evaluation in this section, but it is included in the eight case studies shown in Appendix B, along with the other four models.

For each of the 119 samples, we generate one description using each of the above models, and ask three domain experts to manually score the descriptions generated. When presented to the domain expert, the descriptions from different models are shuffled in a random order for each sample. A score of 2 is given if the description matches the original time series, a score of 1 is given if the description is partially correct, and a score of zero is given if the description is not correct. We sum the scores from all human evaluators for all test (holdout) samples and normalize it to $0 - 1$ to produce the final score for each model. The results are summarized in Figure 2.

As we see, both of our models, Vision (3 epochs) and Vision (1 epoch), significantly outperforms the original LLaVA model. Additionally, training for more epochs seems to lead to a better performance. In fact, using the vision encoder trained for three epochs can lead to a performance that is competitive to GPT-4, although the latter requires first preprocessing the time series using heuristics and statistical tools. Notably, Vision (3 epochs) outperforms GPT-4 on the holdout datasets. We hypothesize that this is because the holdout datasets contain more datasets with complicated seasonalities than the test datasets. Even though Engineering GPT-4 has access to the extracted features, it essentially still performs zero-shot inference. In comparison, our model is finetuned using the proposed TS-Insights dataset and can better leverage the abundance of labeled samples.

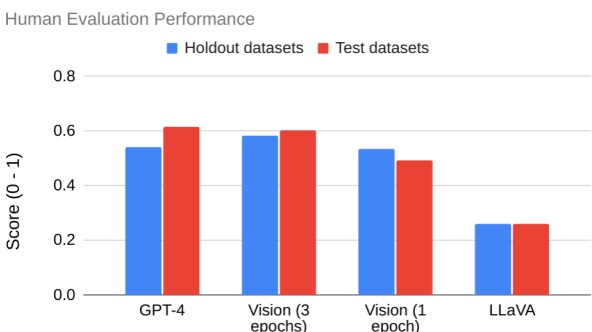

Figure 2: Description evaluation of different models by domain experts.

## 5  Discussions

This work presents the first large dataset with 100k training samples for general time series analysis in the form of time series and natural language pairs. We show that the proposed dataset can enable existing large multimodal models to align time series data with textual descriptions and perform detailed analysis.

In addition to the models evaluated in Section 4, we also tried to use OneFitsAll [5] as the time-series encoder to replace the vision encoder in LLaVA. Our initial attempt shows that using a time-series encoder causes the model to fail to generate coherent descriptions for most samples, which is likely

due to that unlike the original vision encoder, the time-series encoder is not pretrained. Therefore, we leave the pretraining of the time-series encoder as future work. It will be interesting to see whether the proposed dataset can enable large multimodal models to improve forecasting or classification accuracies, since the generated dataset allows them to associate the raw time series vector with common statistical concepts in the form of natural languages.

In terms of the dataset itself, our workflow for generating trend descriptions sheds the light on how descriptions regarding other time series properties can be generated, e.g., the change in volatility, or outlier identification using the extracted residuals. A more challenging task will be to generate descriptions for time series with multiple features, such as by studying their cross-correlations [17].

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

# A  Trend Dataset Details

The 20 datasets involved in generating the TS-Insights dataset are listed below.

| Dataset Name | Granularity | Number of Samples |
|---|---|---|
| saugeenday_dataset | daily | 201 |
| rideshare_dataset_without_missing_values | hourly | 1001 |
| pedestrian_counts_dataset | hourly | 752 |
| oikolab_weather_dataset | hourly | 1141 |
| nn5_daily_dataset_without_missing_values | daily | 301 |
| | tridaily | 51 |
| | weekly | 51 |
| m1_yearly_dataset | yearly | 100 |
| m1_quarterly_dataset | quarterly | 121 |
| m1_monthly_dataset | monthly | 351 |
| london_smart_meters_dataset_without_missing_values | half-hourly | 1000 |
| kdd_cup_2018_dataset_without_missing_values | hourly | 800 |
| kaggle_web_traffic_weekly_dataset | weekly | 800 |
| kaggle_web_traffic_dataset_without_missing_values | daily | 800 |
| hospital_dataset | monthly | 500 |
| fred_md_dataset | monthly | 201 |
| elecdemand_dataset | half-hourly | 102 |
| | hourly | 102 |
| | two-hourly | 80 |
| | three-hourly | 76 |
| | four-hourly | 72 |
| | six-hourly | 64 |
| | eight-hourly | 17 |
| | twice-daily | 17 |
| | daily | 9 |
| covid_mobility_dataset_without_missing_values | daily | 318 |
| covid_deaths_dataset | daily | 280 |
| cif_2016_dataset | monthly | 76 |
| bitcoin_dataset_without_missing_values | daily | 376 |
| australian_electricity_demand_dataset | half-hourly | 600 |
| | **Total** | 10360 |

The following augmentations maybe applied to a given time-series window each with a probability of 50%:

- Jittering: Adding a Gaussian noise to the original time series, where the standard deviation of the Gaussian noise is set to be the standard deviation from a local rolling window of size 4.
- Scaling: Multiplying the original time series with a constant.
- Shifting: Adding a constant to the original time series.
- Smoothing: Convolving the original time series window with an average kernel of a randomly sampled size.
- Downsampling: Only keeping every other $k$ steps, where $k$ is another randomly sampled integer.

Note that multiple augmentations can be applied to get the final augmented window.

The holdout datasets are Electricity Demand (hourly, three-hourly, six-hourly, weekly), M3 (monthly, quarterly, other), M4 (hourly, daily, weekly, monthly, quarterly), Traffic (hourly, bi-hourly, four-hourly), and Weather (daily).

# B  Case Studies

# Hold Out Set – Electricity Hourly Data

- The electricity dataset represents the electricity consumption of 370 clients recorded in 15-minutes periods in Kilowatt (kW) from 2011 to 2014.

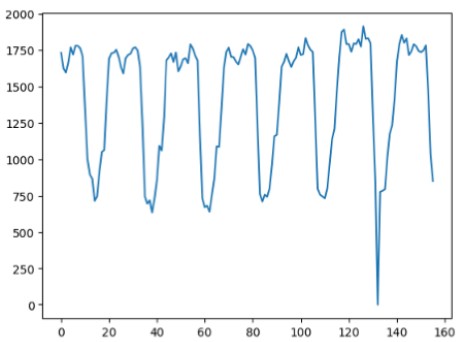

**Engineering GPT:** The trend in the data shows a period of stability, followed by a decline, a return to stability, an increase, and then another period of stability at a higher level.

**Vision 3 epochs:** The trend in the time series starts off steady, then decreases, hits a low point, and eventually increases again.

**Vision 1 epoch:** The trend in the time series starts off stable, then decreases, reaches a low point, and finally increases again.

**LLaVA:** The time-series data shows a gradual upward trend in the beginning, followed by a more pronounced increase and subsequent leveling off.

**GPT:** The trend shows a consistent pattern of rising and falling, with a noticeable drop towards the end.

# Hold Out Set – m3 Other Data

- This dataset contains 174 time series used in the M3 forecasting competition other than the yearly, monthly and quarterly time series.

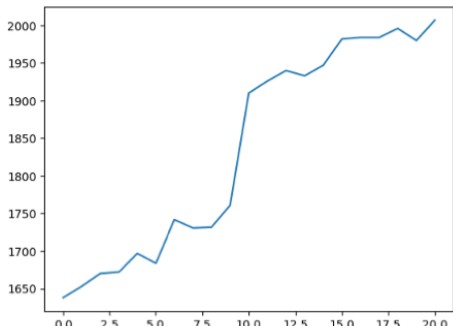

**Engineering GPT :** The data series is constant.

**Vision 3 epochs:** The time series data consistently shows a gradual increase over time.

**Vision 1 epoch:** The time series data shows a gradual upward trend.

**LLaVA:** The time-series data shows a gradual increase in the trend, with a steady growth in the value over time.

**GPT:** The trend shows a general increase over time with a more significant jump towards the end.

## Hold Out Set – m4 Weekly Data

- This dataset contains 359 weekly time series used in the M4 forecasting competition.

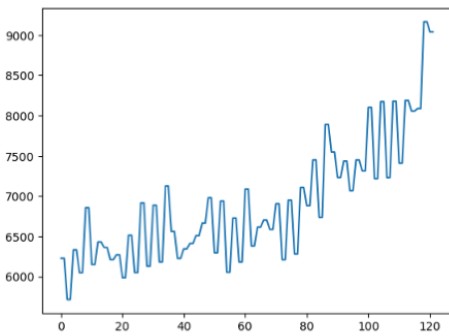

**Engineering GPT:** The trend in the time-series data shows a gradual increase over time.

**Vision 3 epochs:** The trend in the time series starts off steady, then slowly goes up, and finally speeds up.

**Vision 1 epoch:** The trend in the time series data is characterized by a gradual increase.

**LLaVA:** The trend in the time series data shows a gradual increase from the beginning to the middle, followed by a slight decrease and a subsequent increase towards the end.

**GPT:** The trend in the data shows a general increase over time.

## Hold Out Set – Weather

- This dataset contains 3010 daily time series representing the variations of four weather variables: rain, mintemp, maxtemp and solar radiation, measured at the weather stations in Australia.

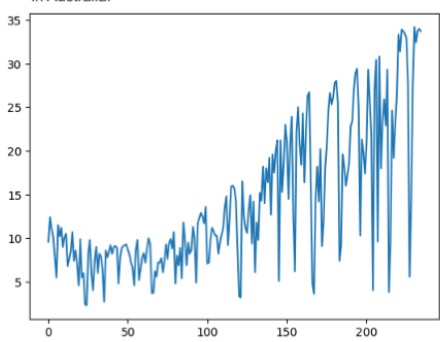

**Engineering GPT:** The trend initially stable, then decreases to a standstill, before gradually increasing at an accelerating rate.

**Vision 3 epochs:** The trend in the time series data shows a gradual increase, followed by a period of stability, and then a more rapid increase.

**Vision 1 epoch:** The trend in the time series data indicates a gradual increase over time.

**LLaVA :** The time-series data shows a gradual increase in the trend over time, with a steady upward slope.

**GPT:** The trend in the data shows a general increase over time, with some periods of stability and occasional decreases.

## Test Set – Australian Electricity Demand

- This dataset contains five time series representing the half hourly electricity demand of five states in Australia: Victoria, New South Wales, Queensland, Tasmania and South Australia.

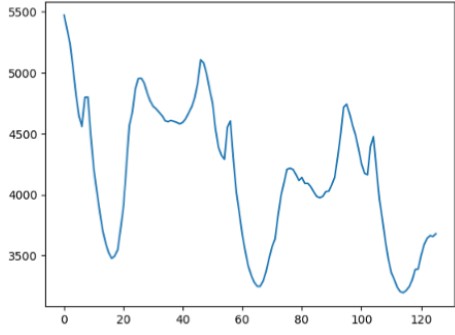

**Engineering GPT:** The trend shows a consistent gradual decrease over time.

**Vision 3 epochs:** The trend gradually decreases over time.

**Vision 1 epoch:** The trend in the time series starts with a decrease, then remains constant, and finally decreases again.

**LLaVA:** The trend in the time series data shows a gradual increase in the value over time, with a slight dip in the middle, and then a steady rise towards the end.

**GPT:** The trend initially decreases, then stabilizes, followed by a slight increase, and finally decreases again.

## Test Set – Covid Deaths Data

- This dataset contains 266 daily time series that represent the COVID-19 deaths in a set of countries and states from 22/01/2020 to 20/08/2020.

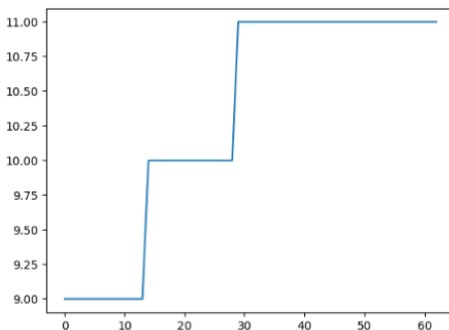

**Engineering GPT:** The trend initially shows a steady increase, then plateaus at a center point.

**Vision 3 epochs:** The time series data shows a period of stability, followed by a sharp increase, and then a period of no change.

**Vision 1 epoch:** The time series starts off with a stable trend, then it suddenly rises and remains at a higher level.

**LLaVA:** The trend in the time series data initially increases, then decreases, and finally stabilizes at a lower level.

**GPT:** The trend in the data shows a steady increase over time.

## Test Set – Pedestrian Counts

- This dataset contains hourly pedestrian counts captured from 66 sensors in Melbourne city starting from May 2009.

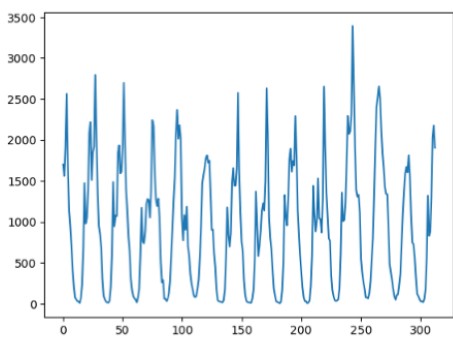

**Engineering GPT:** The trend initially remains stable, then decreases to a low point, before rising sharply and finally declining again.

**Vision 3 epochs:** The trend in the time series starts off stable, then decreases, stabilizes again, increases, and finally stabilizes once more.

**Vision 1 epoch:** The trend in the time series starts off stable, then decreases, stabilizes again, increases, stabilizes once more, and finally decreases.

**LLaVA:** The time-series data shows a gradual increase in the trend, with steady growth over time.

**GPT:** The trend in the data shows a repeated pattern of a sharp increase followed by a gradual decrease.

## Test Set – Saugeenday

- This dataset contains a single very long time series representing the daily mean flow of the Saugeen River at Walkerton in cubic meters per second from 01/01/1915 to 31/12/1979.

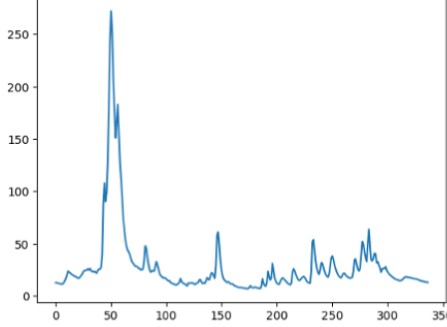

**Engineering GPT:** The trend in the time series starts off stable, then decreases, stabilizes again, increases, stabilizes once more, and finally decreases.

**Vision 3 epochs:** The trend initially increases sharply, then decreases, followed by a period of stability, a slight increase, and finally a decrease.

**Vision 1 epoch:** The trend in the time series starts with a sharp increase, then a sharp decrease, followed by a period of stability, a slight increase, and finally a decrease.

**LLaVA:** The trend in the time series starts with a low value, increases steadily, and then drops back down to a lower value, indicating a gradual upward trend followed by a decline.

**GPT:** The trend initially decreases, then increases sharply, followed by a significant decrease, a slight increase, another decrease, and finally a gradual increase.

