# OpenReview forum: "Insight Miner: A Time Series Analysis Dataset for Cross-Domain Alignment with Natural Language"
_NeurIPS.cc/2023/Workshop/AI4Science — NeurIPS2023-AI4Science Poster_

### Official Review · Reviewer_YkNK · 2023-10-24
**Well written and interesting problem formulation**

**Rating:** 7
**Confidence:** 4

**Review:**

The paper introduces a large-scale multimodal model "Insight Miner," designed to facilitate the generation of comprehensive and domain-specific descriptions from time-series data. This is particularly relevant in fields such as environmental analysis, agriculture, transportation, and finance. The paper presents a new dataset, TS-Insights, which contains pairs of time series data and textual insights, and it is based on a wide range of domains and granularities. The authors preprocess this dataset using a combination of heuristics and statistical tools and then employ GPT-4 for generating coherent trend descriptions based on extracted features. Through instruct tuning with the TS-Insights dataset, Insight Miner outperforms other state-of-the-art multimodal models, including LLaVA and GPT-4, in generating time series descriptions and insights. The paper's findings suggest that leveraging Large Multimodal Models (LMMs) for time series analysis holds promise and could streamline insight mining in scientific domains.

The paper is effectively written and concise, for a workshop submission. It notably provides a valuable, publicly accessible dataset for future research in the field. However, a potential shortcoming lies in the limited depth of experimentation and the absence of automated quantitative performance metrics.

---

### Meta-Review · Area_Chair_XqY1 · 2023-10-26

**Recommendation:** Accept (Poster)
**Confidence:** 3

**Metareview:**

In this paper, authors have primarily created a large-scale multimodel dataset, called TS-Insights, to be released on acceptance, consisting of time-series and corresponding descriptions from different domains. Then, a mutimodel model, called Insight Miner, is developed by finetuning an existing model using TS-Insights dataset. Authors claim the dataset to be first such large-scale dataset where time-series is aligned with language embeddings.
As the reviewer noted, the experiments do not present any interesting results, and the paper does not present anything new also. However, I hope that this dataset will be useful for further research to drawing insights from time-series across domains.  So, I am recommending acceptance (borderline case).